# High-Frequency Ultrasound to Assess Activity in Connective Tissue Panniculitis

**DOI:** 10.3390/jcm10194516

**Published:** 2021-09-29

**Authors:** Priscila Giavedoni, Sebastian Podlipnik, Irene Fuertes de Vega, Pilar Iranzo, José Manuel Mascaró

**Affiliations:** 1Department of Dermatology, Institut Clínic de Medicina i Dermatologia, Hospital Clínic de Barcelona, 08036 Barcelona, Spain; spodlipnik@gmail.com (S.P.); ifuertes@clinic.cat (I.F.d.V.); piranzo@clinic.cat (P.I.); jmmascaro_galy@ub.edu (J.M.M.J.); 2Medical School, University of Barcelona, 08036 Barcelona, Spain; 3Institut d’Investigacions Biomèdiques August Pi i Sunyer (IDIBAPS), 08036 Barcelona, Spain

**Keywords:** high-frequency ultrasound, connective tissue panniculitis, cutaneous lupus erythematosus, dermatomyositis, diagnosis, inflammation

## Abstract

Determining disease activity from clinical signs in patients with connective tissue panniculitis (CTP) is often challenging but is essential for therapeutic decision making, which largely relies on immunosuppressant treatment. High-frequency ultrasound (HFUS) may be useful in supporting such decisions by accurately determining CTP activity. This study aimed to investigate the accuracy of HFUS in identifying signs of CTP activity or inactivity and assess its usefulness in therapeutic decision making. A prospective cohort study of consecutive patients with biopsy-proven CTP receiving HFUS was conducted in a tertiary university hospital (2016–2020). HFUS was performed at inclusion and at each 3- or 6-month follow-up visit, depending on disease activity. Twenty-three patients with CTP were included, and 134 HFUSs were performed. In 59.7% (80) of the evaluations, the clinical presentation did not show whether CTP was active or not. In these cases, HFUS showed activity in 38.7% (31) and inactivity in 61.3% (49). In 71.25% (57) of the visits, HFUS was the determinant for therapeutic decisions. Further follow-up showed consistent clinical and HFUS responses in all unclear cases after treatment modification. HFUS appears to be a useful adjunct to the clinical examination for CTP to assess activity and make therapeutic decisions.

## 1. Introduction

Connective tissue panniculitis (CTP) is a rare cutaneous manifestation of connective tissue diseases associated with lupus erythematosus (LE) and dermatomyositis (DM) [1,2]. CTP usually presents as warm, painful erythematous nodules or plaques [3,4,5]. Clinical manifestations are not always specific enough to determine whether CTP is active. Although skin biopsy may be useful [3,6], it is invasive, may result in scarring, and is not adequate for follow-up [7]. CTP does not respond to topical therapies and is usually treated with antimalarials, systemic corticosteroids, and immunosuppressants [8]. Establishing whether the disease is active is fundamental to therapeutic decision making.

High-frequency ultrasound (HFUS) is increasingly used in dermatology [9,10,11]. In inflammatory diseases, such as morphea and graft-versus-host disease [12], it may determine whether lesions are active or healing [13]. HFUS has been shown to be helpful in the diagnosis of panniculitis, differentiating between septal and lobular forms [14,15,16].

The main objective of this study was to determine the usefulness of HFUS in assessing inflammatory activity in patients with CTP and whether it aids therapeutic decision making.

## 2. Materials and Methods

### 2.1. Design

A prospective cohort study was carried out between January 2016 and December 2020 at the Department of Dermatology, Hospital Clinic of Barcelona.

Patients with confirmed lupus or dermatomyositis with clinical and histological signs of panniculitis with ≥2 clinical and ultrasound findings were included. Patients without a confirmatory biopsy were omitted, even if they had typical clinical signs of panniculitis. Patients who underwent only an ultrasound study without subsequent follow-up were also excluded.

### 2.2. Patients

All consecutive patients diagnosed with histologically confirmed CTP attended at our department. Patients with systemic LE (SLE) fulfilled American 2019 European League Against Rheumatism/American College of Rheumatology classification criteria for systemic lupus erythematosus [17]. The diagnosis of cutaneous lupus erythematosus was clinical, and a compatible skin biopsy was performed in all patients. Patients with DM met the criteria proposed at the 119th ENMC International Workshop, 2003, Naarden, The Netherlands [18].

Panniculitis was considered clinically active when lesions presented as warm, painful erythematous nodules or plaques. If lesions showed only atrophic, painless plaques without signs of inflammation, they were considered inactive. Those that did not meet defined criteria of activity or inactivity were considered as “clinically unspecified activity”.

### 2.3. Skin Ultrasound

HFUS was performed by a dermatologist trained in the diagnosis and management of autoimmune diseases and HFUS. The dermatologist who performed HFUS trained with Dr. Worstman and Dr. Alfageme, recognized experts in the field. In addition, the dermatologist performs HFUS with a specific schedule of 20 patients per week and has six years of experience, with an average of more than 400 skin ultrasounds per year, so is considered an expert according to the international group of specialists in the field [19].

HFUS was performed by a dermatologist trained in the diagnosis and management of autoimmune diseases and HFUS. Esaote MyLab™Class C equipment (Genova headquarters, Via E. Melen, 77; 16152 Genoa, Italy) was used with high-frequency probes between 10, 18, and 22 MHz. The software was based on Windows^®^ XP (Redmond; Washington, DC, USA) and used DICOM services (Radiological Society of North America; Oak Brook, IL, USA) to download worklists and save acquired ultrasound images on a network storage CD-R, DVD, or removable device connected to a USB, and print images on a network copy device. HFUS was performed using both B-mode gray-scale ultrasound imaging and color Doppler mode. Color Doppler mode was used to determine the direction of blood flow and power Doppler mode to assess skin vascularization, as it is more sensitive to low-flow/low-speed vessels found in inflammatory skin lesions. Spectral Doppler was used to distinguish veins from arteries and, in the latter, to determine the systolic peak, diastolic peak, and resistance index. Doppler measurements were made with an insonation angle ≤60°. The pulse repetition frequency was fixed at 750. The color gain was variable and adjusted to the value immediately below the noise threshold [20].

Clinical and ultrasound follow-up was made every 3 months if there was persistent CTP inflammatory activity. In patients with clinical and ultrasound inactivity in two consecutive assessments, follow-up visits were made at 6 to 12 months. Therapeutic management was decided according to clinical and ultrasound findings each visit. Some patients reported discomfort, a slight increase in temperature, and edema at panniculitis sites, but these symptoms were mild and could not be confirmed by clinical signs. Therefore, these cases were considered as “nonspecific skin symptoms”.

In patients with nonspecific symptoms, therapeutic decisions were made according to ultrasound findings. In patients with established symptoms, treatment was decided according to the clinical findings and clinical–ultrasound concordance was evaluated.

The therapeutic decision was classified as: 1. treatment intensification, including increasing the dose or starting a new drug; 2. reduction in treatment, when the dose of ≥1 of the drugs used was reduced; 3. treatment discontinuation, when the drugs used were discontinued; and 4. no change in treatment. In all patients, the therapeutic decision taken at the first visit was evaluated according to the clinical and ultrasound evolution at the three-month follow-up.

Blood flow is rarely detected in the normal dermis on color Doppler by current equipment, and isolated vessels <1 mm are identified in the hypodermis [21]. Arterial vessels in healthy skin have a low velocity with a systolic peak of <10 cm/s on spectral Doppler [21]. Based on studies on the usefulness of ultrasound in determining panniculitis-causing inflammatory disease activity, the following cut-off values were used to determine negative Doppler: isolated vessels in the dermis and hypodermis, with a systolic peak of <10 cm/s and a resistance index of <0.7 [12,13]. Active panniculitis was defined as fulfillment of ≥2 of the 3 criteria in Doppler mode: systolic peak > 10 cm/s, resistance index > 0.7 MHz, and vessel diameter > 1 mm.

With the probes and equipment used, in healthy skin, the epidermis is observed as a hyperechogenic line of up to 0.1 mm, the normal dermis is usually seen as a hyperechogenic band to the hypodermis, and the hypodermis is composed of large hypoechoic areas corresponding to the lobules, with thin hyperechoic bands corresponding to the septa [21] (Figure 1A).

The variables analyzed in HFUS B-mode considered indicative of active disease were: hyperechogenicity of lobes in all cases with or without hypoechogenicity and thickening of the hypodermis septa [12,13,14,21]. Increased or decreased echogenicity were considered when compared with healthy perilesional skin and contralateral skin septa were considered thickened when ≥1 mm in ≥3 septa [14].

Other signs present in active disease were hypoechogenicity of the dermis and loss of the dermo-hypodermal line [12,21]. Affected areas were compared with perilesional or contralateral healthy skin to determine if echogenicity was increased or reduced.

### 2.4. Statistical Analyses

Pearson’s X^2^ test and trend test for ordinal variables were used to compare categorical and ordinal variables, respectively. For continuous variables, the Wilcoxon test was used for comparison between two groups of samples and the Kruskal–Wallis test to compare multiple groups. The analyses were undertaken using the computing environment R and RStudio and used a two-sided type I error of 0.05.

## 3. Results

### 3.1. Baseline and Clinical Characteristics

We included 23 patients with CTP (Table 1), on whom 134 HFUSs were performed. The median age was 44 years (IQR, 18–78), and 22 (95.6%) were women. Eighteen patients (78.3%) had LE, and five (21.7%) had DM. Associated autoimmune diseases included: optic neuritis (*n* = 1; 4.3%), autoimmune hepatitis (*n* = 1; 4.3%), alopecia areata (*n* = 1; 4.3%), and rheumatoid arthritis (*n* = 1; 4.3%).

In patients with LE, nine (50%) had panniculitis without other clinical disease manifestations. Of the remaining LE patients, six (33%) had SLE, five (27.8%) had discoid LE, and one (5.5%) had subacute LE (Table 1). CTP involved two or more locations in 15 (65%) of the cases, and the most frequently affected sites were the arms (*n* = 15; 65.2%) and scalp (*n* = 8; 34.8%). All patients received antimalarial drugs during the disease evolution: hydroxychloroquine, chloroquine, and/or mepacrine. Most required immunosuppressive drugs; the most frequently used was oral prednisone at doses of more than 15 mg/day for more than 1 month in 52% of patients. Other drugs used were methotrexate (*n* = 8; 34.8%), mycophenolate mofetil (*n* = 7; 30.4%), belimumab (*n* = 2; 8.7%), and IV immunoglobulins (*n* = 2; 8.7%). Ruxolitinib, cyclosporine, infliximab, tofacitinib, and rituximab were used in only one patient.

### 3.2. Patient Assessment Three Months after Diagnosis

Of the 23 patients, 18 presented nonspecific clinical activity at the first visit: HFUS showed no inflammation in 7 patients (38.9%), and inflammation in color Doppler mode in 11 (61.1%). Based on the HFUS findings, in the seven patients without inflammation, treatment reduced in four (57.3%): the dose of prednisone decreased in three cases and that of hydroxychloroquine in one; in one case, hydroxychloroquine was discontinued, and in two patients no changes were made, both of whom were receiving only topical corticosteroids on demand. In these patients with nonspecific clinical activity and without HFUS activity, HFUS findings aided dose reductions in immunosuppressive drugs in three (42.9%) patients. At 3 months, three (42.9%) cases showed clinical inactivity, four (57.1%) persisted with nonspecific clinical manifestations, and six out of seven patients (85.7%) had ultrasound inactivity. Only one patient (14.3%) showed ultrasound activity and unspecified clinical activity (Figure 2).

In the 11 patients with unclear disease activity on clinical assessment in whom HFUS showed CTP activity, treatment was initiated in 9 (81.8%) patients: hydroxychloroquine was started in 7 patients, methotrexate in 1, and prednisone in 1 patient; in 1 patient, the dose of methotrexate was increased; and no changes were made in 1 case. At three months, two patients had clinical and ultrasound inactivity, unspecified clinical activity persisted in three (27.3%) but with inactive ultrasound, and five patients with unspecified clinical activity had active ultrasound. In one case, the clinical activity became active, and the ultrasound remained active.

### 3.3. Clinical Ultrasound Follow-Up during the Complete Study Period

A total of 134 HFUSs were performed on the 23 patients during a follow-up of 4 years. HFUS had an excellent correlation with clinical findings (Table 2), both in cases where signs of inflammation were observed and where scarring was seen. In all clinically active lesions (*n* = 14; 10.4%), color Doppler showed activity in at least two of the three parameters: peak systolic > 10 cm/s, resistance index > 0.7 MHz, and vessel diameter > 1 mm. In clinically inactive lesions (*n* = 40; 29.8%), the Doppler mode of HFUS showed inactivity in 39 (97.5%) cases. In one case, clinical features were inactive but the Doppler mode showed activity.

### 3.4. The Usefulness of Ultrasound in the Presence of Nonspecific Clinical Findings

In 80 evaluations (59.7%), the clinical presentation was not sufficiently specific to determine whether the disease was active. In these cases, ultrasound showed 31 (38.7%) had activity and 49 (61.3%) did not. In 57 (71.2%) assessments, HFUS was decisive in modifying treatment: treatment was reduced in 29 assessments (50.9%), and increased in 28 (40.1%) (Table 3).

### 3.5. Ultrasound Features of Inflammation

Of the 134 HFUSs performed, 46 (34.3%) showed active CTP in Doppler mode. All patients with active HFUS panniculitis had at least two of the three criteria for Doppler activity. In addition, 23 HFUSs (50%) showed all three activity criteria in Doppler mode. A total of 45 HFUSs (97.8%) had a systolic peak >10 cm/s, 38 HFUSs (82.6%) had a resistance index >0.7, and 33 HFUSs (71.7%) had vessels >1 mm in diameter. The 13 HFUSs (28.3%) with active CTP but with vessels <1 mm in diameter had lesions on the scalp. HFUSs that were active in color Doppler mode showed hyperechogenicity of lobules, hypoechogenicity, and thickening of septa in 45 (97.8%), 21 (45.6%), and 9 (19.5%) cases, respectively. Other B-mode findings found in active HFUS in the Doppler mode were: hypoechogenicity of the dermis in 41 HFUSs (89.1%), and loss of dermo-hypodermal differentiation in 38 (82.6%). Figure 1A shows the color Doppler ultrasonographic features of active panniculitis. Atrophy of the subcutaneous tissue was identified in 114 HFUSs (85%) (Figure 1B), and calcinosis cutis in 14 (10.4%) (Figure 1C).

## 4. Discussion

Our study is the first to show that HFUS can be used to assess inflammatory activity in patients with CTP and support therapeutic decisions, especially when disease activity is unclear on clinical assessment. All patients presenting with inflammation on clinical assessment had increased flow in color Doppler mode. In two-thirds of the evaluations, the clinical assessment was nonspecific. In almost half of these patients, HFUS was the determinant in deciding to initiate/advance treatment; in the remaining evaluations, treatment was reduced/suspended since no activity was observed on HFUS. In one patient who appeared inactive on physical evaluation, HFUS showed increased Doppler flow.

In daily clinical practice, simultaneous HFUS and clinical evaluation strengthens non-invasive and rapid decision making. We suggest HFUS may be useful for an objective assessment of inflammatory activity. The chief advantages of HFUS include that it is inexpensive and non-invasive, does not use ionizing radiation, and is broadly available [21]. Expert dermatologists have the advantage of being able to make an excellent clinical–ultrasonographic correlation [14]. Currently, HFUS is increasingly used in dermatology both for diagnosis and follow-up of inflammatory diseases and tumors, and cosmetic dermatology [22,23]. In Spain, HFUS is routinely used in clinical practice in many tertiary hospitals [24,25].

HFUS may be as helpful as pathology in guiding the clinical diagnosis of active CTP. Studies comparing HFUS findings with histology found a high intra-observer correlation [14]. When a biopsy is needed to confirm the diagnosis, HFUS can aid the choice of the most inflamed sites for sample collection, as is performed in other organs, such as the liver, breast, or lymph nodes [26,27]. The main weakness of HFUS is that it is operator-dependent, and expertise requires previous training [19,28]. HFUS appears to be helpful in the follow-up of sclerosing autoimmune diseases, such as morphea and systemic sclerosis. It assists in determining the activity of these diseases with a notable impact on clinical management [29,30]. In systemic sclerosis, an excellent correlation of HFUS findings with dermal collagen content has been demonstrated. In addition, HFUS allows the detection of subclinical abnormalities that would allow early treatment [31].

CTP HFUS findings are well-defined in B-mode and color Doppler [14]. In color Doppler mode, the systolic peak, resistance index, and the diameter of inflamed vessels can be measured [32]. In other skin diseases causing panniculitis, such as morphea or sclerodermiform graft-versus-host disease, color Doppler mode values have been established to define inflammation [12,13]. Color Doppler allows noninvasive quantification of inflammation and objective evaluation of the treatment response at each follow-up visit simultaneously with the clinical evaluation.

Our results also suggest that HFUS could avoid diagnostic biopsies in patients who meet the clinical criteria for LE or DM and who present with clinical signs of CTP. Moreover, HFUS appears to be potentially useful in increasing early diagnoses in cases with subtle clinical manifestations. In our series, patients with subtle clinical CTP manifestations had well-defined inflammation on HFUS. The diagnosis of CTP based on clinical manifestations alone is often delayed for several years during which time, in addition to pain and inflammation, atrophy and calcium deposits in the subcutaneous tissue may develop [8].

HFUS can be used to quantify subcutaneous tissue atrophy [21,33], one of the frequent sequelae in CTP [5]. In cases of esthetic restorative treatment, it would allow evaluation of the results [34]. In addition, it allows measurement of the size and depth of calcium deposits, enabling assessment of long-term treatment response [35]. One of our patients had extensive plaques of calcium deposits of ≥20 cm in length on both thighs, making it impossible to follow them objectively by physical examination.

Computed tomography (CT) and magnetic resonance imaging (MRI) can help diagnose CTP and associated complications such as atrophy and calcinosis cutis [16]. In Parry–Romberg syndrome, the usefulness of CT and MRI in measuring subcutaneous tissue atrophy was described [36]. However, these techniques are usually not as accessible as HFUS, are more expensive, uncomfortable for the patient, and may require the use of a contrast medium and, in the case of CT, ionizing radiation [32].

The main limitation of our study is that HFUS findings of inflammation or panniculitis inactivity were not compared with histology. Therefore, we had to consider panniculitis as active when previously described criteria for inflammatory panniculitis were met. In addition, the study was performed in a single center with only one person carrying out HFUS and therefore inter-observer variability was not assessed. Although the study did not include many patients, panniculitis is an uncommon cutaneous manifestation of LE and is even more unusual with DM.

## 5. Conclusions

HFUS appears to be useful in enhancing the accurate determination of disease activity and supporting therapeutic decision-making in patients with CTP. Studies assessing the accuracy of HFUS in evaluating CTP inflammatory activity and examining its positive and negative predictive values are warranted.

## Figures and Tables

**Figure 1 jcm-10-04516-f001:**
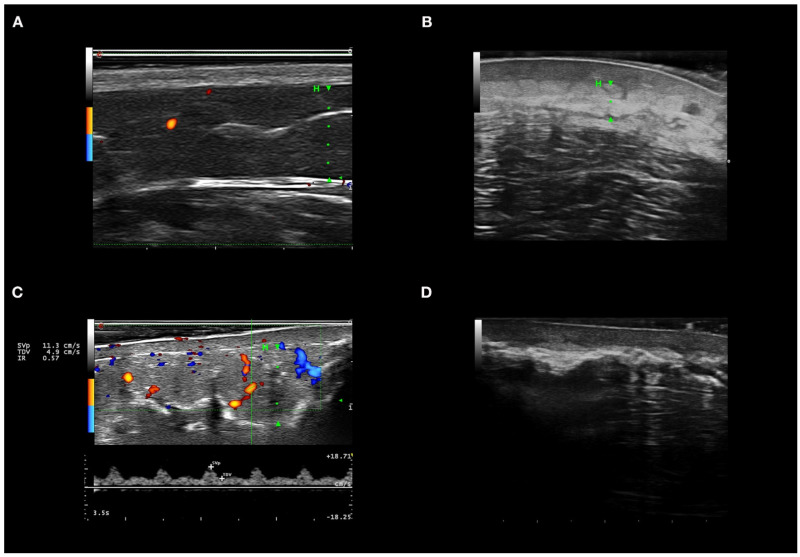
High-frequency ultrasound in autoimmune panniculitis. (**A**) Healthy skin in color Doppler mode. (**B**) Inflammation of dermis and hypodermis; the dermis is hypoechoic, and the hypodermis with hyperechoic lobules in B mode. (**C**) Panniculitis with hyperechogenicity of lobules and marked increase in flow in color Doppler mode. (**D**) Extensive calcium deposits with posterior acoustic shadowing. In images A, B, and C, the hypodermis is marked with an H and a green line. In Figure 1D, the calcium deposits do not allow evaluation of the hypodermis as they produce a posterior acoustic shadow.

**Figure 2 jcm-10-04516-f002:**
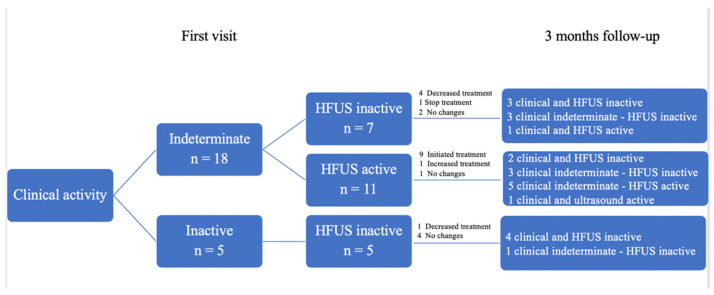
Usefulness of HFUS in therapeutic decision-making at first visit and 3-month follow-up. HFUS: High-frequency Ultrasound.

**Table 1 jcm-10-04516-t001:** Clinical characteristics of patients with connective tissue panniculitis.

	Overall (*N* = 23)
Age (years)	
Median	44 (18–78)
Sex	
Female	22 (95.7%)
Male	1 (4.3%)
Autoimmune disease	
Lupus erythematosus *	18 (78.3)
Isolated lupus panniculitis	9 (50%)
Systemic lupus erythematosus	6 (33.3%)
Chronic discoid lupus	5 (27.8%)
Subacute cutaneous lupus	1 (5.5%)
Dermatomyositis	5 (21.7%)
Classic dermatomyositis	3 (60%)
Amyopathic dermatomyositis	2 (40%)
Other cutaneous manifestations associated with lupus	
Cicatricial alopecia	3 (16.7%)
Perniosis	2 (11.2%)
Oral ulcers	1 (5.6%)
Other systemic manifestations associated with lupus	
Arthritis	4 (22.2%)
Arthralgia	2 (11.2%)
Nephritis	2 (11.2%)
Serositis	1 (5.6%)
Cytopenia	1 (5.6%)
Myositis	1 (5.6%)
Other cutaneous manifestations associated with dermatomyositis	
Calcinosis	4 (80%)
Gottron papules	4 (80%)
Heliotrope erythema	3 (60%)
Periungual telangiectasias	3 (60%)
Cutaneous ulcers	2 (40%)
V-shaped erythema at the chest	2 (40%)
Erythema on proximal thighs	2 (40%)
Facial erythema	1 (20%)
Other systemic manifestations associated with dermatomyositis	
Myositis	3 (60%)
Interstitial lung disease	1 (20%)
Panniculitis location	
Arms	15 (65.2%)
Scalp	8 (34.8%)
Thighs	6 (26.1%)
Buttocks	6 (26.1%)
Face	3 (13%)
Breasts	3 (13%)
Trunk	3 (13%)
Forearms	3 (13%)
Legs	2 (8.7%)
Number of systemic treatments, mean (range)	3 (1 -17)
Number of ultrasounds performed per patient	
Median (range)	3 (2–13)

* Three patients had systemic lupus erythematosus and chronic discoid lupus.

**Table 2 jcm-10-04516-t002:** Baseline HFUS characteristics of patients with connective tissue panniculitis.

	Clinical Active (*N* = 14)	Clinical Inactive (*N* = 40)	Total (*N* = 54)	*p* Value	Sensitivity	Specificity
Ultrasound activity				<0.001	100%	97.50%
Active	14 (100.0%)	1 (2.5%)	15 (27.8%)			
Inactive	0 (0.0%)	39 (97.5%)	39 (72.2%)			
Hypoechoic dermis				<0.001	78.57%	92.50%
Yes	11 (78.6%)	3 (7.5%)	14 (25.9%)			
No	3 (21.4%)	37 (92.5%)	40 (74.1%)			
Dermo-hypodermic limit				<0.001	78.57%	75%
Undefined	11 (78.6%)	10 (25.0%)	21 (38.9%)			
Defined	3 (21.4%)	30 (75.0%)	33 (61.1%)			
Hypoechoic septa				<0.001	42.86%	97.50%
Yes	6 (42.9%)	1 (2.5%)	7 (13.0%)			
No	8 (57.1%)	39 (97.5%)	47 (87.0%)			
Thickened septa				0.003	21.43%	100%
Yes	3 (21.4%)	0 (0.0%)	3 (5.6%)			
No	11 (78.6%)	40 (100.0%)	51 (94.4%)			
Hyperechoic lobules				<0.001	92.86%	72.50%
Yes	13 (92.9%)	11 (27.5%)	24 (44.4%)			
No	1 (7.1%)	29 (72.5%)	30 (55.6%)			
Vessel diameter >1 mm				<0.001	85.71%	100%
Yes	12 (85.7%)	0 (0.0%)	12 (22.2%)			
No	2 (14.3%)	40 (100.0%)	42 (77.8%)			
Peak systolic flow >10 cm/s				<0.001	100%	97.50%
Yes	14 (100.0%)	1 (2.5%)	15 (27.8%)			
No	0 (0.0%)	39 (97.5%)	39 (72.2%)			
Resistive index >0.7				<0.001	64.29%	97.50%
Yes	9 (64.3%)	1 (2.5%)	10 (18.5%)			
No	5 (35.7%)	39 (97.5%)	44 (81.5%)			
Calcinosis cutis				0.347	21.43%	65%
Yes	3 (21.4%)	14 (35.0%)	17 (31.5%)			
No	11 (78.6%)	26 (65.0%)	37 (68.5%)			
Subcutaneous tissue atrophy				0.348	92.86%	17.50%
Yes	13 (92.9%)	33 (82.5%)	46 (85.2%)			
No	1 (7.1%)	7 (17.5%)	8 (14.8%)			

**Table 3 jcm-10-04516-t003:** Therapeutic modifications supported by high-resolution Doppler ultrasound results in patients with undetermined clinical findings.

Undetermined Clinical Activity	80 (100%)
Therapeutic changes based on HFUS	57 (71.2%)
Decrease/Stop treatment	29 (36.2%)
stop prednisone	1 (3.4%)
stop hydroxychloroquine	1 (3.4%)
decrease prednisone	17 (58.6%)
decrease hydroxychloroquine	5 (17.2%)
decrease ruxolitinib	2 (6.9%)
decrease mepacrine	1 (3.4%)
decrease methotrexate	1 (3.4%)
Increase treatment	28 (35%)
start hydroxychloroquine	8 (28.6%)
start prednisone	2 (7.1)
start methotrexate	1 (3.6%)
start mepacrine	1 (3.6%)
start mycophenolate	1 (3.6%)
start hydroxychloroquine	1 (3.6%)
start belimumab	1 (3.6%)
start tofacitinib	1 (3.6%)
increase prednisone	4 (14.3%)
increase hydroxychloroquine	3 (10.7%)
increase methotrexate	3 (10.7%)
increase mycophenolate	1 (3.6%)

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
