# Peer review of "High-Frequency Ultrasound to Assess Activity in Connective Tissue Panniculitis"

_jcm, 2021, doi:10.3390/jcm10194516_

Round 1
Reviewer 1 Report
Thank you for allowing me to review this topic. It is of great interest to me. The literature on this topic is scant, yet the assessment and management of panniculitis is something we do frequently, thus supporting the relevance and utility of this work. These findings have the potential to change the way we approach the assessment of panniculitis in patients who have had confirmatory biopsies. As mentioned in the paper, the measurements are based on previously used parameters for other skin diseases, so it will be helpful to have a prospective study in CTP to assess and validate these parameters.
HFUS has been used in Dermatology to assess disease activity in a number of other diseases, including morphea and scleroderma. There are many more publications on the use of HFUS in these conditions. I think it would be helpful to know how HFUS is incorporated into the assessment of those diseases and how your approach is either similar or different.
One of the limitations mentioned is the availability and skill-level of the sonographer, which is very important. It should also be stated how the dermatologist learned to perform HFUS skin evaluations and how long they have been practicing this technique. It is nice that you lay out objective parameters, so that in the future, CTP should more easily assessable for the more naive sonographer.
In figure 2, the legend is not clear. You need to add an "A" and "D" (they are there not labeled). For someone who is not familiar with the different skin layers, it would be helpful to mark the subcutaneous layer, so we know what we are looking at, AND your point would be better illustrated if you included some "normal" (either perilesional or contralateral/unaffected location) HFUS scans. The representation calcinosis cutis in "C" is not clear - this could be better demonstrated sonographically (and before/after pictures would be helpful to see how you can monitor over time, if that is what you are demonstrating here). I don't know that the panoramic view offers much added benefit/information.
This is a very nicely done project, I think it just needs some grammatical edits and more supportive/explanatory information (For example, on page 2, line 77-78, describing "nonspecific skin symptoms" would be helpful - this was mentioned earlier in the text as well - so could be addressed there; on the same page, line 97, "hyperechogenicity of lobes" was stated, but the meaning of this may be unclear to the non-dermatologist, so that needs to be better explained/described.)
Finally, it should be mentioned why you used the ACR criteria for SLE and Peters and Bohan criteria for DM, when there are more updated classification criteria published.
Overall a great work, and I look forward to reading more about this!
Author Response
Point 1: Thank you for allowing me to review this topic. It is of great interest to me. The literature on this topic is scant, yet the assessment and management of panniculitis is something we do frequently, thus supporting the relevance and utility of this work. These findings have the potential to change the way we approach the assessment of panniculitis in patients who have had confirmatory biopsies. As mentioned in the paper, the measurements are based on previously used parameters for other skin diseases, so it will be helpful to have a prospective study in CTP to assess and validate these parameters.

Response 1: Thank you very much for your comments; we also find it essential to perform prospective multicenter studies to validate our results.
Point 2: HFUS has been used in Dermatology to assess disease activity in a number of other diseases, including morphea and scleroderma. There are many more publications on the use of HFUS in these conditions. I think it would be helpful to know how HFUS is incorporated into the assessment of those diseases and how your approach is either similar or different.
Response 2: In morphea and systemic sclerosis, HFUS has been shown to effectively assess lesions activity and guide therapeutic decisions. In addition, ultrasound in systemic sclerosis has been shown to correlate well with histological findings and detect disease at early stages.
We have added these utilities of HFUS, with the respective references from lines 302 - 307.
Point 3: One of the limitations mentioned is the availability and skill-level of the sonographer, which is very important. It should also be stated how the dermatologist learned to perform HFUS skin evaluations and how long they have been practicing this technique. It is nice that you lay out objective parameters, so that in the future, CTP should more easily assessable for the more naive sonographer.
Response 3: The dermatologist who performed HFUS trained with Dr. Worstman and Dr. Alfageme recognized experts in the field. In addition, he performs HFUS with a specific schedule of 20 patients per week and has six years of experience, with an average of more than 400 skin ultrasounds per year. Considered an expert according to the international group of specialists in the field. This information is aggregated in lines 95 - 99.
Point 4: In figure 2, the legend is not clear. You need to add an "A" and "D" (they are there not labeled). For someone who is not familiar with the different skin layers, it would be helpful to mark the subcutaneous layer, so we know what we are looking at, AND your point would be better illustrated if you included some "normal" (either perilesional or contralateral/unaffected location) HFUS scans. The representation calcinosis cutis in "C" is not clear - this could be better demonstrated sonographically (and before/after pictures would be helpful to see how you can monitor over time, if that is what you are demonstrating here). I don't know that the panoramic view offers much added benefit/information.
Response 4: In Figure 2, we have corrected the legend with the corresponding letters and identified the dermis and hypodermis in each case. A now corresponds to healthy skin in color Doppler mode. B: inflammation of dermis and hypodermis; the dermis is hypoechoic, and the hypodermis with hyperechoic lobules in B mode. C: Panniculitis with hyperechogenicity of lobules and marked increase of flow in color Doppler mode. D: extensive calcium deposits with posterior acoustic shadowing. In images A, B, and C, we have marked the hypodermis with an "H" and a green line. In Figure 2D, the calcium deposits do not allow evaluation of the hypodermis as they produce a posterior acoustic shadow. This information is aggregated in lines 242 - 248.
Point 5: This is a very nicely done project, I think it just needs some grammatical edits and more supportive/explanatory information (For example, on page 2, line 77-78, describing "nonspecific skin symptoms" would be helpful - this was mentioned earlier in the text as well - so could be addressed there; on the same page, line 97, "hyperechogenicity of lobes" was stated, but the meaning of this may be unclear to the non-dermatologist, so that needs to be better explained/described.)
Response 5: Some patients reported discomfort, a slight increase in temperature, and edema at panniculitis sites, but these symptoms were mild and could not be confirmed by clinical signs. Therefore, these cases were considered as "nonspecific skin symptoms." This information is aggregated in lines 115 – 118; We have added the image and the ultrasound description of healthy skin so that the hyperechogenicity of the hypodermis in panniculitis can be easily interpreted. With the probes and equipment used, in healthy skin, the epidermis is observed as a hyperechogenic line of up to 0.1 mm, the normal dermis is usually seen as a hyperechogenic band to the hypodermis, and the hypodermis is composed of large hypoechoic areas corresponding to the lobules, with thin hyperechoic bands corresponding to the septa. (Figure 2-A). 137 - 141
Point 6: Finally, it should be mentioned why you used the ACR criteria for SLE and Peters and Bohan criteria for DM, when there are more updated classification criteria published.
Response 6: We used the classic ACR criteria for SLE and those of Peters and Bohan for DM. However, we reviewed our data, and patients with systemic SLE met the classification criteria for systemic lupus erythematosus of the European League Against Rheumatism and the American College of Rheumatology. The diagnosis of cutaneous lupus erythematosus was clinical, and a compatible skin biopsy was performed in all patients. Furthermore, patients with DM met the criteria proposed at the 119th ENMC International Workshop, 2003, Naarden, The Netherlands[18]. We have therefore modified with more updated references the criteria for lupus and dermatomyositis. This information is aggregated in lines 83 – 88.

Reviewer 2 Report
Many hospital do not have autoimmune ultrasound trained physician, how do your study help them?
Did you check inflammatory markers (ESR & CRP) during this study? Did they correlate with your ultrasound findings?
Author Response
Point 1: Many hospital do not have autoimmune ultrasound trained physician, how do your study help them?
Response 1: We believe that ultrasound is a tool that is being widely used. In Spain, most of the hospitals that train dermatology residents have an ultrasound in their routine practice. Sonography use in dermatology is spreading to many other countries. In national and international congresses, cutaneous ultrasound has its place. Cutaneous ultrasound equipment is affordable compared to other technologies used in medicine, such as tomography, MRI, or confocal microscopy. In addition, a short training period is needed to be able to analyze images in dermatology.
All this makes us think that it is crucial to disseminate ultrasound findings in dermatological pathologies and the contribution that ultrasound can have to follow-up and therapeutic decision-making in dermatology.
Point 2: Did you check inflammatory markers (ESR & CRP) during this study? Did they correlate with your ultrasound findings?
Response 2: It is a very interesting comment. Unfortunately, analytical assessments were not performed in most patients, so we have not included them in the results. We plan to do a prospective multicenter study to validate our results, and it would be essential to include inflammatory markers to assess whether there is a correlation with clinical and ultrasound findings. Thank you very much for your comment; it seems very important to us.
